# Optimal dosing of dihydroartemisinin-piperaquine for seasonal malaria chemoprevention in young children

Palang Chotsiri[1], Issaka Zongo[2], Paul Milligan [3], Yves Daniel Compaore [2], Anyirékun Fabrice Somé[2], Daniel Chandramohan[4], Warunee Hanpithakpong[1], François Nosten [5,6], Brian Greenwood[4], Philip J. Rosenthal[7], Nicholas J. White[1,5], Jean-Bosco Ouédraogo[2] & Joel Tarning [1,5]

Young children are the population most severely affected by *Plasmodium falciparum* malaria. Seasonal malaria chemoprevention (SMC) with amodiaquine and sulfadoxine-pyrimethamine provides substantial benefit to this vulnerable population, but resistance to the drugs will develop. Here, we evaluate the use of dihydroartemisinin-piperaquine as an alternative regimen in 179 children (aged 2.33–58.1 months). Allometrically scaled body weight on pharmacokinetic parameters of piperaquine result in lower drug exposures in small children after a standard mg per kg dosage. A covariate-free sigmoidal $E_{MAX}$-model describes the interval to malaria re-infections satisfactorily. Population-based simulations suggest that small children would benefit from a higher dosage according to the WHO 2015 guideline. Increasing the dihydroartemisinin-piperaquine dosage and extending the dose schedule to four monthly doses result in a predicted relative reduction in malaria incidence of up to 58% during the high transmission season. The higher and extended dosing schedule to cover the high transmission period for SMC could improve the preventive efficacy substantially.

[1] Faculty of Tropical Medicine, Department of Clinical Pharmacology, Mahidol-Oxford Tropical Medicine Research Unit, Mahidol University, Bangkok 10400, Thailand. [2] Institut de Recherche en Sciences de la Santé, Bobo-Dioulasso, Burkina Faso. [3] Department of Epidemiology and Population Health, London School of Hygiene and Tropical Medicine, London WC1E 7HT, United Kingdom. [4] Department of Disease Control, London School of Hygiene and Tropical Medicine, London WC1E 7HT, United Kingdom. [5] Centre for Tropical Medicine and Global Health, Nuffield Department of Clinical Medicine, University of Oxford, London OX3 7LJ, United Kingdom. [6] Faculty of Tropical Medicine, Shoklo Malaria Research Unit, Mahidol-Oxford Tropical Medicine Research Unit, Mahidol University, Mae Sot 63110, Thailand. [7] Department of Medicine, University of California, Box 0811 San Francisco CA 94143 CA, USA. Correspondence and requests for materials should be addressed to J.T. (email: joel@tropmedres.ac)

Malaria is one of the most important infectious diseases in humans. In areas of intense malaria transmission children, who have yet to develop protective immunity, are especially vulnerable to malaria. In 2017, ~61% of worldwide malaria-related mortality occurred in children below the age of 5 years[1]. Intermittent preventive treatment of malaria consists of a full antimalarial treatment regimen given at regular intervals in order to eliminate any existing infections and to produce sufficiently high residual drug concentrations to prevent new infections until the next dose is given or the need diminishes. Intermittent preventive treatment of malaria has been shown to be beneficial for high-risk populations, such as pregnant women in sub-Saharan Africa[2–4]. In addition, seasonal malaria chemoprevention (SMC), i.e. intermittent preventive treatment during the high-transmission season, is recommended for children living in regions of high yet seasonal malaria transmission, principally the Sahelian regions of West Africa. Intermittent preventive therapy in pregnancy relies on sulfadoxine-pyrimethamine (SP) and SMC on SP plus amodiaquine. In the sub-Sahel areas where SMC is being used, SP plus amodiaquine currently remains effective but resistance is likely to develop[5]. In parts of SE Africa, where malaria is highly seasonal and SMC could potentially be used, SMC is not recommended because of resistance to SP. Dihydroartemisinin-piperaquine (DHA-PQ) is a potential alternative regimen for SMC in young children in areas where there is resistance to SP[3,4,6].

The fixed-dose antimalarial drug combination of oral DHA-PQ is an attractive alternative for SMC. It can be administered once daily for three days. A large pooled analysis of 12 different studies in 6 countries between 2006 and 2009 showed excellent efficacy (98.7% at day 28) and safety (4.8% total incidence of early vomiting) in the treatment of uncomplicated *Plasmodium falciparum* malaria[7]. The pharmacokinetic properties of DHA-PQ have been described in various patient groups and in healthy volunteers[8–10]. Dihydroartemisinin is a rapidly eliminated but potent artemisinin derivative that kills a large proportion of the infecting malaria parasite biomass. Three days of treatment exposes two asexual cycles in *P. falciparum* infections to the drug. In contrast, piperaquine is eliminated slowly with a long terminal elimination half-life (20–30 days). Piperaquine is responsible for killing the residual parasites in the body, which remain after three days of dihydroartemisinin and thereby prevents recrudescent malaria. The long terminal elimination half-life of piperaquine also provides a long post-treatment prophylactic effect, which lasts at least four weeks if adequate doses are given and fully sensitive parasites are prevalent. Piperaquine is eliminated primarily by metabolism by cytochrome P450 3A4 (CYP3A4)[11,12]. Piperaquine is a highly lipophilic molecule and studies have shown that the absorption can increase with up to 121% (95% CI: 26-216%) if administered together with a high fat meal[13,14]. However, administration with a low fat meal demonstrated no increased absorption[15,16].

The pharmacokinetic and pharmacodynamic characteristics of DHA-PQ support its use in SMC. A study from Uganda reported prophylactic efficacies of monthly DHA-PQ chemoprevention in children of 31% (95% CI: 6–49%), 67% (95% CI: 54–76%), and 97% (95% CI: 89–99%) in the low, median, and high piperaquine exposure groups, respectively[17]. A full 3-day treatment regimen of DHA-PQ given monthly for 9 months was well tolerated and resulted in a high protection rate of 98% (95% CI: 96–99%) against *P. falciparum* malaria in adults at high risk of malaria on the Thailand-Myanmar border[18,19]. Chemoprevention in young children receiving monthly DHA-PQ resulted in a three year protective efficacy of 58% (95% CI: 7–44%), and was superior to both monthly SP and daily trimethoprim-sulfamethoxazole[20]. This protective efficacy was lower than expected because of poor drug adherence. On the other hand, a chemoprevention study in schoolchildren receiving monthly supervised DHA-PQ showed a remarkably high protective effect, i.e. reduced malaria incidence by 96% (95% CI: 88–99%) and reduced asymptomatic malaria by 94% (95% CI: 19–56%)[21]. Preventive efficacy of DHA-PQ has been shown in Southeast Asia in the treatment of *P. vivax* malaria, where the drug combination eliminated blood stage parasites rapidly and suppressed a large proportion of the first relapses, which emerge usually three weeks after the acute infection[22]. When used in the treatment of young children with acute uncomplicated *P. falciparum* malaria, the original manufacturer recommended treatment regimen provides sub-optimal plasma concentrations of piperaquine[10,23]. Limited information is available on the pharmacokinetic and pharmacodynamic properties of DHA-PQ when used for malaria prevention in young children.

The primary aim of this study was to investigate the pharmacokinetic and pharmacodynamic properties of piperaquine in SMC for young children in Burkina Faso and thereby inform optimum dosing. The final pharmacokinetic–pharmacodynamic model was used to simulate protective effects against malaria with different treatment regimens.

## Results
**Study demographic and data**. This study included a total of 741 children who received oral DHA-PQ in a clinical trial [6] of SMC (Table 1). The nested pharmacokinetic–pharmacodynamic (PKPD) group included 179 children and the pharmacodynamic (PD) group included 562 children. Only the children in the PKPD group provided blood samples for drug measurements. The pharmacokinetic and pharmacodynamic model was developed based on patients in the PKPD group, and the data from the children in the PD group were used as an external validation of the final pharmacodynamic time-to-event model. Of the 741 apparently healthy children enroled in the study, 42.1% (312) had a positive blood smear for *P. falciparum* malaria at the beginning of the study, and 21.9% (162) and 36.9% (274) of children acquired at least one new malaria episode during the three rounds of monthly SMC and the 2 months of follow-up, respectively. The median time to new infections was 90 days post-enrolment (range: 13–153 days). Twenty-two children (3.0%) were lost to follow-up. Children who received SMC with SP and amodiaquine in the original trial had a slightly lower incidence of malaria (19.5%) during the three months of active treatment[6].

A total of 26, 82, and 40 children provided capillary plasma samples (466 capillary samples) and 6, 10, and 15 children provided venous plasma samples (71 venous samples) during the first, second, and third study months, respectively. Only 1 of 466 capillary plasma samples and 3 out of 71 venous plasma samples were measured to be below the lower limit of quantification (LLOQ) for piperaquine, and were omitted from the subsequent pharmacokinetic–pharmacodynamic analysis. Non-linear mixed-effects modelling was performed in order to characterise the pharmacokinetic and pharmacodynamic properties of piperaquine.

**Pharmacokinetic model**. The pharmacokinetic model was developed based on the data from capillary and venous plasma concentrations of the children in the PKPD group ($n = 179$). Due to the sparsity of measured drug concentrations, a frequentist prior approach was used in order to stabilize the estimation of typical pharmacokinetic parameters and inter-individual variability. Observed capillary plasma piperaquine concentration-time data were described accurately by a two-transit-compartment absorption model, followed by a three compartment distribution

**Table 1 Clinical study demographics, sample collection, and treatment outcomes**

| Parameter | PKPD group (primary analysis) | PD group (external validation) | Total |
|---|---|---|---|
| Total no. of children | 179 | 562 | 741 |
| Total no. of samples (capillary/venous plasma) | 466/71 | NA | NA |
| Total monthly dose of piperaquine base (mg kg$^{-1}$) | 29.2 (18.0–39.0) | 29.7 (16.7–55.4) | 29.7 (16.7–55.4) |
| Total monthly dose of dihydroartemisinin (mg kg$^{-1}$) | 6.32 (3.90–8.45) | 6.43 (3.61–12.0) | 6.43 (3.61–12.0) |
| Continuous and categorical covariates at admission | | | |
| Age (months) | 32.1 (2.33–58.1) | 24.0 (3.00–59.3) | 26.1 (2.33–59.3) |
| Body weight (kg) | 11.0 (4.20–18.3) | 10.5 (5.00–21.0) | 10.6 (4.20–21.0) |
| Axillary temperature at admission (ºC) | 36.7 (35.0–39.3) | 36.7 (35.0–40.4) | 36.7 (35.0–40.4) |
| Number of patients with malaria (%) | 71 (39.6%) | 250 (44.5%) | 312 (42.1%) |
| Number of male patients (%) | 93 (51.9%) | 277 (49.3%) | 370 (49.9%) |
| Treatment outcomes during follow-up | | | |
| Number of patients with malaria (%) | 110 (61.4%) | 322 (57.3%) | 432 (58.2%) |
| Time-to malaria (days) | 107 (28–149) | 90 (13–153) | 90 (13–153) |
| Parasitaemia in patients with malaria (parasites µL$^{-1}$) | 48,926 (64–1,496,212) | 36,081 (12–260,000) | 39,275 (12–1,496,212) |
| Number of patients lost before day 90 (%) | 4 (2.19%) | 18 (3.20%) | 22 (2.95%) |
| Follow-up time of lost patients (days) | 60 (60–62) | 60.5 (20–80) | 60 (20–89) |

Data from the children in the PKPD group were used to develop the pharmacokinetic and pharmacodynamic model, and the data from the children in the PD group (no blood samples collected) were used for external validation of the final pharmacodynamic model. All values are given as median (range) unless otherwise indicated
*PK* pharmacokinetics, *PD* pharmacodynamics, *NA* not available

model as reported previously[10]. Venous and capillary piperaquine plasma concentrations were modelled simultaneously by estimating a venous-capillary conversion factor at a population level (illustrated in the Supplementary Figure 1). The present study showed a significantly lower exposure to piperaquine compared to the prior model. This was, therefore, corrected for by implementing a categorical covariate on the relative bioavailability. Including an enzyme maturation effect on oral clearance during the first two years of age improved the model fit significantly, but resulted in an unrealistic and poorly estimated enzyme maturation half-life (TM$_{50}$) of less than one month. Therefore, the maturation effect was omitted in the final pharmacokinetic model. No other covariates were significant in the stepwise addition and elimination covariate approach. Final pharmacokinetic parameter estimates and relative standard errors are presented in the Table 2 and secondary pharmacokinetic parameter estimates are presented in Supplementary Table 1.

The final model described the observed concentration-time profiles accurately with no major model misspecification (Supplementary Figure 2), and with good predictive performance (Fig. 1a). A numerical predictive check resulted in 3.8% (95% CI: 3.00–7.52%) and 6.4% (95% CI: 2.82–7.33%) of the observed data being above and below the simulated 90% prediction interval, respectively. Eta and epsilon shrinkages were relatively high in the final model because of the sparseness of the observed data: CL/F = 54%, V$_C$/F = 46%, Q$_2$/F = 74%, V$_{P2}$/F = 65%, MTT = 76%, F = 21%, and σ = 26.8%. Bootstrapping the final pharmacokinetic model showed robust parameter estimates with acceptable relative standard errors.

**Pharmacodynamic model.** Overall, 58.5% of children (436/741) presented with malaria during the 4 months study period (3 rounds of monthly SMC and 2 months of passive follow-up). Median time to recurrent malaria was 90 days (range: 13–153 days) after starting the study, corresponding to approximately one month after the last study treatment. The children in the PKPD group had a slightly higher malaria incidence compared to those in the PD group (62.3% vs. 57.3%, respectively).

An interval-censoring time-to-event model was applied successfully to the children in the PKPD group, using parasitaemia-corrected back-extrapolation to the time of emergence from the pre-erythocytic liver stage infections (illustrated in the supplementary Figure 3). This model was suggested by Bergstrand et al.[19] and it provides a more mechanistic understanding of recurrent malaria infections. A constant baseline hazard model with a sigmoidal $E_{MAX}$ antimalarial effect relationship of piperaquine concentrations and study outcome was superior to other models. The model also contained a fixed dihydroartemisinin effect, reducing the hazard of reinfection to zero during each treatment occasion because of the very potent parasite elimination effect of dihydroartemisinin. No covariate effects on the pharmacodynamic parameters were found using a stepwise covariate approach. Time-to-malaria infection in the PD group was used as an external validation of the final pharmacokinetic–pharmacodynamic model. A visual predictive check of the time-to-event model showed good predictive performance, both for the internal (Fig. 1b and Supplementary Figure 4) and external (Fig. 1c and Supplementary Figure 5) validation. Bootstrapping showed robust parameter estimates with acceptable relative standard errors.

The minimum inhibitory concentration (MIC) of piperaquine was estimated in these children based on the predicted piperaquine concentrations at the start of novel blood stage infections, using the same back-extrapolation methodology described above. The upper (95$^{th}$) percentile of these drug concentrations is assumed to be the highest possible concentration which still allows parasite replication. Thus, the predicted MIC values in these patients (i.e. the 95$^{th}$ percentile of piperaquine venous plasma concentrations at the start of blood stage infection) were between 12.9 ng mL$^{-1}$ and 17.5 ng mL$^{-1}$ (equivalent to capillary blood concentrations of 33.9 ng mL$^{-1}$ and 45.5 ng mL$^{-1}$, respectively) based on the start and the end of the likely time period of new parasites emerging from the liver (illustrated in the Supplementary Figure 6).

**Dosing regimen simulations.** A high proportion (33.3%, 6/28 and 60.9%, 84/138) of observed day 7 piperaquine venous and capillary plasma concentrations were below a threshold for therapeutic success of 30 ng mL$^{-1}$ and 57 ng mL$^{-1}$, respectively, previously defined for the treatment falciparum malaria[10,24].

**Table 2 Parameter estimates from the final pharmacokinetic–pharmacodynamic model of piperaquine in children receiving seasonal malaria chemoprevention in Burkina Faso**

| Parameters[a] | Prior estimates[b] | Population estimates[c] | 95% confidence interval[d] | %RSE[d] |
|---|---|---|---|---|
| **Pharmacokinetics** | | | | |
| MTT (h) | 2.15 | 1.37 | 0.506–1.93 | 26.9 |
| CL/F (L h$^{-1}$) | 7.50 | 7.36 | 7.52–7.84 | 1.04 |
| $V_C$/F (L) | 247 | 314 | 282–356 | 5.80 |
| $Q_1$/F (L h$^{-1}$) | 13.1 | 9.78 | 6.89–12.7 | 15.1 |
| $V_{P1}$/F (L h$^{-1}$) | 254 | 274 | 266–284 | 1.69 |
| $Q_2$/F (L) | 10.8 | 10.8 | 10.5–11.1 | 1.30 |
| $V_{P2}$/F (L h$^{-1}$) | 3340 | 3490 | 3410–3580 | 1.30 |
| Conversion$_{CAP-VEN}$ | — | 0.380 | 0.313–0.450 | 8.99 |
| $\sigma_{CP}$ | — | 0.305 | 0.256–0.346 | 7.76 |
| $\sigma_{VP}$ | — | 0.666 | 0.489–0.797 | 11.9 |
| **Covariates** | | | | |
| Relative F | — | 0.726 | 0.675–0.781 | 3.71 |
| **Inter-individual variability (%CV)** | | | | |
| MTT (h) | 0.494 (79.9) | 0.574 (88.1) | 0.440–0.827 | 9.35 |
| CL/F (L h$^{-1}$) | 0.0433 (21.0) | 0.0438 (21.2) | 0.0362–0.0540 | 5.30 |
| $V_C$/F (L) | — | 0.665 (97.2) | 0.0825–1.14 | 18.8 |
| $Q_2$/F (L) | 0.0487 (22.3) | 0.0478 (22.1) | 0.0444–0.0531 | 2.37 |
| $V_{P2}$/F (L h$^{-1}$) | — | 0.0486 (22.3) | 0.00000486–0.283 | 65.0 |
| F | 0.0735 (27.6) | 0.114 (34.7) | 0.0805–0.164 | 9.40 |
| **Pharmacodynamics** | | | | |
| BASE (year$^{-1}$) | — | 6.28 | 5.13–11.2 | 9.35 |
| $IC_{50}$ (ng mL$^{-1}$) | — | 3.66 | 2.09–5.40 | 15.1 |
| $\gamma$ | — | 1.79 | 1.12–2.45 | 12.5 |

[a]*BASE* baseline hazard, *CL* elimination clearance, *Conversion$_{CAP-VEN}$* proportional conversion factor between capillary and venous drug measurements, *F* relative bioavailability, *γ* shape factor, *$IC_{50}$* piperaquine venous plasma concentrations associated with a reduction of the baseline hazard by 50%, *MTT* mean absorption transit time, *Q* intercompartment clearance, *$\sigma_{CP}$* variance of proportional residual error of the capillary samples, *$\sigma_{VP}$* variance of proportional residual error of the venous samples, *$V_C$* central volume of distribution, *$V_P$* peripheral volume of distribution
[b]The final model and parameter estimates from the pharmacokinetic study of piperaquine in children[10] were used as prior parameter estimates
[c]Computed population mean parameter estimates from NONMEM were calculated for a typical child of 18.0 kg body weight. The coefficient of variation (%CV) for inter-individual variability was calculated as $100 \times \sqrt{exp(\omega^2) - 1}$
[d]Computed from the non-parametric bootstrap method of the final pharmacokinetic model ($n = 1000$), and pharmacodynamic model ($n = 500$). The 95% confidence intervals are based on the 2.5th and 97.5th percentile of the bootstrap parameter estimates, and the % relative standard errors (%RSE) are computed as $100 \times$ (standard deviation/mean value)

Thus, the final pharmacokinetic–pharmacodynamic model was used to evaluate different dosing regimens designed to increase drug exposure in this group of children. The protocol regimen, the standard WHO 2010 treatment regimen and the updated increased dose treatment regimen (WHO 2015) were used for simulations (Supplementary Table 2). Simulated concentration-time profiles of the standard WHO 2010 treatment regimen and the increased treatment dose regimen resulted in 75% and 50%, respectively, of simulated day 7 venous plasma concentrations falling below the previously defined threshold of 30 ng mL$^{-1}$ for treatment therapeutic success (Fig. 2)[24]. The predicted cumulative malaria incidences in children of 4–20 kg body weight were estimated to be 33.0% (95% CI: 24.0–43.3%) and 22.5% (95% CI: 17.5–34.5%) at day 60 after a single treatment of the standard WHO 2010 dosing regimen and the increased dosing regimen, respectively. The predicted cumulative malaria incidences in children of 4–20 kg body weight at day 120 were predicted to be 27.0% (95% CI: 14.5–45.0%) and 16.5% (95% CI: 9.00–26.0%) after three months of SMC using the WHO 2010 dosing regimen and the increased dosing regimen, respectively (Figs. 3, 4 and Table 3). However, the predicted cumulative malaria incidences during three months of active SMC (i.e. 0–90 days) were predicted to be 12.0% (95% CI: 4.23–27.0%) and 5.0% (95% CI: 1.50–10.5%) according to the WHO 2010 and WHO 2015 dosing regimens, respectively. Irrespective of the dose regimen, total malaria incidence doubled between day 90 and 120, suggesting that another SMC dose at day 90 is needed (i.e. four months of SMC) in this region. The predicted cumulative malaria incidence at day 120 was 13.0% (95% CI: 0.40–30.0%) and 5.50% (95% CI:

1.50–11.5%) after four months of SMC with the WHO 2010 and WHO 2015 dosing regimens, respectively.

## Discussion

DHA-PQ is a well-tolerated, safe and effective malaria treatment[25]. It is also under evaluation as preventive therapy in high-risk groups, such as young children living in areas of high transmission[4]. Antimalarial treatment dosing for the paediatric population has traditionally been prescribed based on a linear extrapolation from adult dosing[43]. This simple extrapolation fails to take into account the non-linear relationship between body weight and pharmacokinetic parameters determining exposure (i.e. elimination clearance). This has resulted in lower exposures to piperaquine in small children and an increased risk of therapeutic failures, which would be expected to promote the emergence of drug resistant parasites[10,23]. As chemoprevention, monthly DHA-PQ exhibited an excellent protective effect against malaria in children[3,4]. Several studies have characterized the pharmacokinetic properties of piperaquine in African children using both non-compartmental approaches and non-linear mixed-effects modelling[10,27–29]. Beyond extrapolation from the treatment studies, dosing has not been evaluated extensively for SMC.

In this population pharmacokinetic–pharmacodynamic assessment there were not enough sparse capillary and venous plasma concentration measurements (~3–4 samples per individual) to develop a pharmacokinetic model with a good degree of precision. Thus, a frequentist prior approach was applied in order

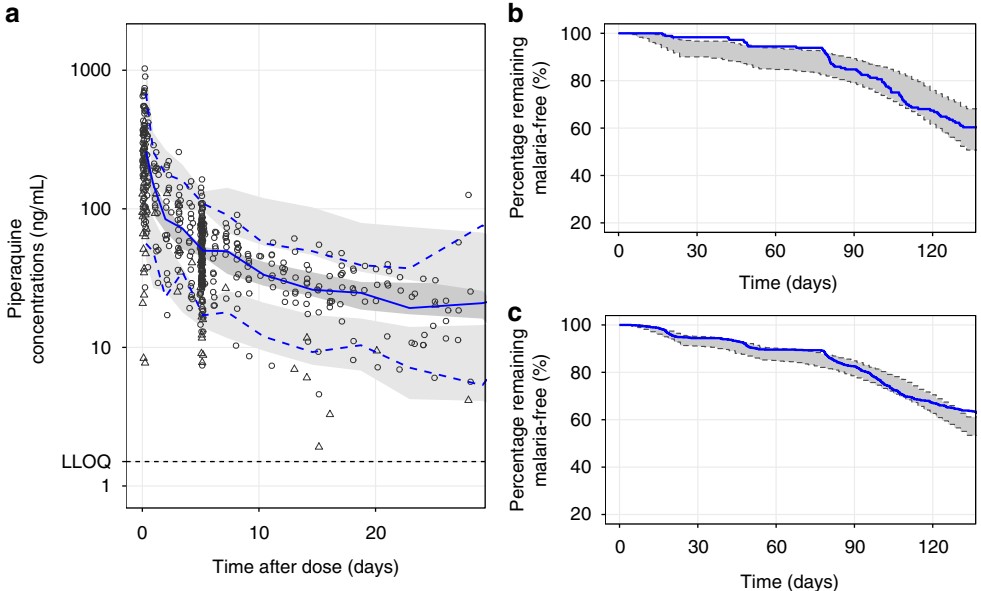

**Fig. 1** Visual predictive checks of the final population pharmacokinetic and pharmacodynamic model of piperaquine. **a** The final population pharmacokinetic model, **b** the interval-censoring time-to-event model of the internal data, and **c** the final pharmacodynamic model predicting the external data. Open circles represent observed capillary plasma piperaquine concentrations and open triangles represent observed venous plasma concentrations. A solid line represents the median observed plasma concentrations and dashed lines represent the 5th and 95th percentiles of the observed plasma concentrations. Shaded areas represent the predicted 95% confidence intervals of each percentile. Solid lines in panel (**b**) and (**c**) represent observed Kaplan–Meier survival plots. Shaded areas represent the 95% prediction intervals

to stabilize the structural model. The prior model was developed on dense data collected from a treatment study in children (2–10 years of age) that was conducted in the same region[10] and should therefore be generalizable to the study population presented here. All pharmacokinetic parameters, inter-individual variability, and uncertainty measurements were included, and the developed model was penalised for model estimates, which were far from prior estimates. The pharmacokinetic parameter estimates were therefore centralised to the median body weight in the prior model and scaled with a fixed allometric function. The prior pharmacokinetic model was a flexible transit-compartment model followed by a three-compartment disposition model, which is in agreement with recent pharmacokinetic publications on piperaquine in children, adults and pregnant women[8,10,22,30].

Almost all venous plasma samples were collected at the same time-points as capillary plasma samples, and results showed strong correlation between capillary and venous concentration measurements (Supplementary Figure 1). However, capillary piperaquine concentrations were estimated to be substantially higher than venous measurements (163%) in the same patient, similar to the linear regression model (Supplementary Figure 1) and to results from previous reports[22,31]. Incorporation of an estimated conversion factor in the population pharmacokinetic model allowed for simultaneous fit of both capillary and venous concentrations, which increased the amount of data and so improved model stability. The conversion factor was somewhat higher than that previously reported in patients with *falciparum* malaria (90%) and *vivax* malaria (41%)[22,31]. The exact reason for higher observed concentrations in capillary plasma compared to venous plasma could not be elucidated from the data in this study, but capillary samples are affected by peripheral perfusion and tissue fluid concentrations as they contain a variable admixture of interstitial fluid. Differences in this estimated conversion factor might result from a relatively small number of

venous samples in this study and also a different population compared to that studied previously (African children vs Asian adults).

A categorical study covariate was necessary to compensate for an unexplained lower body weight-normalised exposure in the present study compared to that in the older children used in the prior model. The data collected in this study were not sufficient to explain this discrepancy. A difference in diet could explain differences in exposure, since piperaquine absorption is increased when it is administered together with a high fat meal[14]. Maturation effects and other possible admission covariates could not explain the lower exposure in this study.

The long terminal elimination half-life of piperaquine (median of 21.3 days, 95% CI: 19.8–25.4 days) resulted in drug accumulation with 23% and 30% higher exposures to piperaquine in treatment periods two and three, respectively, compared to the first treatment period. This was in agreement with the observed data showing increasing concentrations over time, with median day 7 piperaquine capillary plasma concentrations of 35.6, 48.3, and 57.6 ng mL$^{-1}$ after the first, second, and third month of SMC, respectively. The final pharmacokinetic model showed good predictive performance and was therefore deemed suitable to be implemented in the pharmacokinetic–pharmacodynamic model.

Malaria infections on admission and during follow-up were not PCR genotyped, and it was, therefore, not possible to distinguish new from recrudescent infections, and thereby assess treatment outcomes. However, in previous studies from Africa, DHA-PQ has shown excellent treatment efficacy, with very few recrudescent infections after therapy. Malaria infections during the 107 days of follow-up were described successfully with a time-to-event model, assuming a constant baseline hazard. Such time-to-event models have been used previously to describe time-to-malaria re-infections[19,22]. The implemented model estimated a

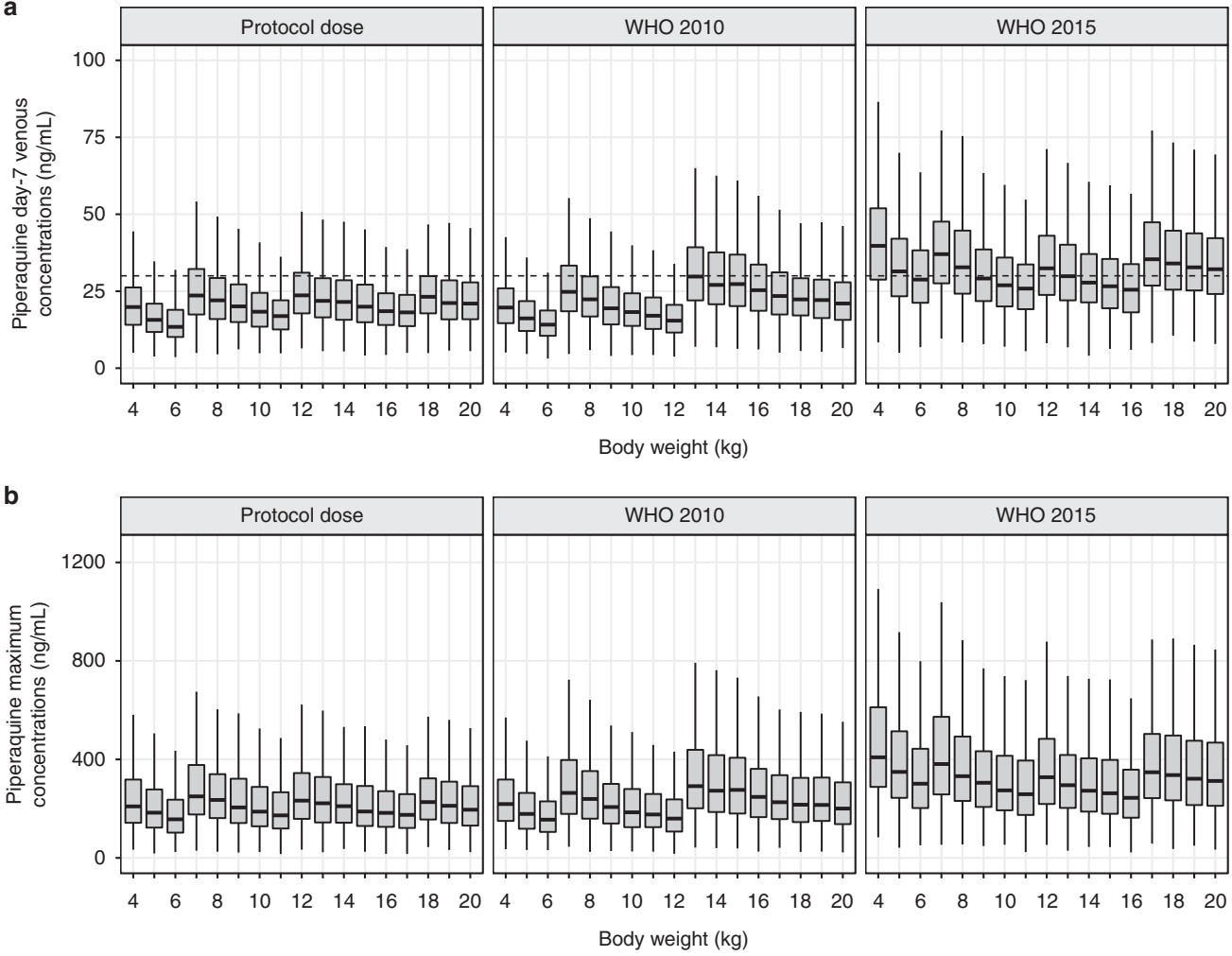

**Fig. 2** Simulated venous piperaquine concentrations. **a** Day-7 piperaquine concentrations and **b** peak piperaquine concentrations after different dosing regimens, stratified by body weight. The box-whisker plots represent the median with inter-quartile range and the 95% prediction interval of 1000 simulated individuals per body weight. The horizontal dashed line represents the previously defined 30 ng mL$^{-1}$ cut-off concentration at day 7 associated with therapeutic success[24]

baseline hazard of 6.28 (95% CI: 5.50–14.2) malaria infections per year. Similarly, the reported incidence rate in children in Burkina Faso was 5.0 (95% CI: 4.8–5.2) infections per year[32]. As the current study was performed during the high malaria transmission season, a higher baseline hazard was expected in the current study. Individually estimated piperaquine concentrations were implemented with a sigmoidal $E_{MAX}$-type model to describe the prevention of new malaria infections during SMC. The estimated piperaquine venous concentration needed to reduce the baseline hazard by 50% (IC$_{50}$) was 3.66 ng mL$^{-1}$ (equivalent to the capillary plasma concentrations of 9.62 ng mL$^{-1}$), which is close to that previously estimated in adults in Thailand before the recent emergence of piperaquine resistance[22]. Observed parasitaemia at the time of malaria detection was used to back-extrapolate a starting time interval of emerging asexual blood stage infections[19]. The parasite multiplication rate and length of parasite life cycle are known to vary between patients, i.e. previously published results present a 90% prediction interval between 5.5-fold and 12.3-fold per 48 hours[33]. Therefore, since the parasite growth rate was not observed in this study, the growth rate was assumed to vary between 5-fold and 10-fold per 48 h. This extrapolated interval-censoring approach should produce more biologically accurate parameter estimates and reflect

the residual piperaquine concentrations needed to prevent the multiplication of parasites emerging from the liver (at a biomass of $10^4$–$10^5$ parasites). However, the observed microscopy detection limit of circulating parasites coincides with the pyrogenic density of approximately $10^8$ parasites[34]. Thus, any sub-microscopic and asymptomatic infections present at the monthly round of SMC were assumed to be completely cleared by the administered drugs, since dihydroartemisinin eliminates ~$10^4$ parasites per 48 h (i.e. a standard 3-day treatment is expected to eliminate all asymptomatic infections)[35,36]. Therefore, the time-to-event model included a categorical dihydroartemisinin drug effect to improve further the mechanistic characterisation of SMC with artemisinin-based combination therapies. A weekly dosing regimen has been suggested to be superior to the monthly regimen in SMC[37]. However, the weekly dosing would not necessarily be reflected by the three-day dosing of dihydroartemisinin component in this model since it assumes that the full 3-day treatment would completely eliminate circulating parasites. A prospective study comparing monthly and weekly SMC is needed in order to establish the optimal dosing strategy in SMC.

The predicted MIC value of piperaquine in these children was estimated based on the likely starting time of blood stage infections. Thus, this value should represent the venous plasma concentration

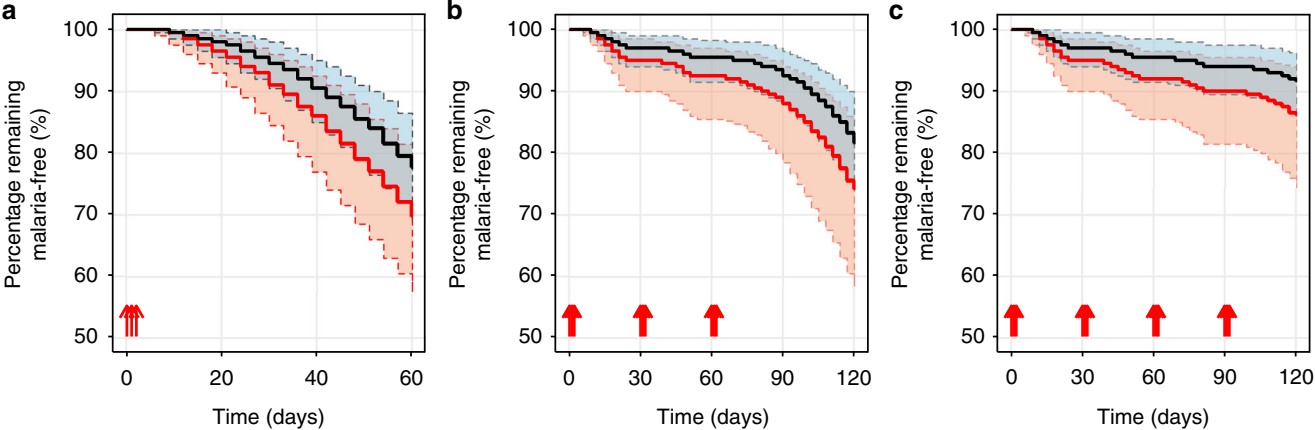

**Fig. 3** Simulation of the expected pharmacodynamic outcome. **a** remaining malaria-free after a single treatment regimen, **b** remaining malaria-free after three rounds of monthly dose regimens (day 0, 30, and 60), and **c** remaining malaria-free after four rounds of monthly dose regimens (day 0, 30, 60, and 90). Red lines represent the standard WHO 2010 dosing regimen[52,43] and black lines the increased dosing regimens according to the revised WHO 2015 recommendation[26] in children (4–20 kg; $n = 200$ individuals per body weight, 100 replications). Solid lines represent the predicted median survival estimate of the Kaplan–Meier plot and shaded areas represent the 95% prediction intervals. Upward red arrows represent the time of DHA-PQ administrations

needed to suppress the parasite multiplication rate to one, and therefore signify a realistic target concentration for preventive chemotherapy. The venous plasma MIC was estimated at 12.9 to 17.5 ng mL$^{-1}$. This predicted MIC range was of similar magnitude to ex-vivo assessments of IC$_{50}$ values for the piperaquine sensitive isolates in this region, i.e. the geometric mean IC$_{50}$ for piperaquine was 20.8 ng mL$^{-1}$ (range: 4.18 to 42.2 ng mL$^{-1}$) in Cameroon[38] and 17.2 ng mL$^{-1}$ (IQR: 9.16 to 24.8 ng mL$^{-1}$) in Kenya. The IC$_{50}$ of piperaquine from *P. falciparum* community isolates in Uganda decreased from 10.2 ng mL$^{-1}$ (95% CI: 9.16–11.5 ng mL$^{-1}$)[39] in 2010–2013 to 4.01 ng mL$^{-1}$ (95% CI: 3.21–4.98 ng mL$^{-1}$) in 2016[40]. For successful chemoprevention, piperaquine concentrations should be maintained above the MIC value in order to eliminate all residual parasites and to prevent new infections arising during the rainy season.

The final pharmacokinetic–pharmacodynamic model illustrated clearly the benefit of SMC during the high malaria transmission season in Burkina Faso. Intermittent preventive DHA-PQ treatment in adult subjects in Thailand demonstrated that the subjects with a monthly trough piperaquine plasma concentration above 31 ng mL$^{-1}$ did not have any malaria episodes[18]. However, using the previously recommended fixed daily dosage target of 18 mg/kg of piperaquine resulted in 33.0% (95% CI: 24.0–43.3) of children having a malaria episode during the 60 days of follow-up period (post-SMC). In updated 2015 guidelines, the WHO recommended an increase in the dosage of DHA-PQ for malaria treatment in young children[26]. Based on large-scale SMC simulations using the final pharmacokinetic–pharmacodynamic model, an increased piperaquine dosage would have reduced the number of children with sub-therapeutic day 7 concentrations from 50% to 25% using the threshold of 30 ng mL$^{-1}$. The simulated peak piperaquine concentrations after an increased dosage were 314 ng mL$^{-1}$ (95% CI: 97.8–1120 ng mL$^{-1}$), which was similar to values obtained in adults treated for malaria[41]. The simulated increased dosage would have reduced the predicted cumulative malaria incidence at day 120 by 38.8% in children at 4–20 kg body weight when administered as three months of SMC. However, most of the predicted malaria incidence occurred between day 90 and 120, suggesting that one additional monthly round of SMC is needed

in this area. The predicted malaria incidence at day 120 after four months of SMC was similar to the incidence at day 90 after three rounds of SMC, indicating that one additional month of SMC could have a substantial impact on the malarial incidence in this region. Thus, we propose that the increased piperaquine dosage recommended for the treatment of small children (WHO 2015 guideline) should be applied to SMC as well. If DHA-PQ is used for SMC, an extended dosing schedule (i.e. four months of SMC) should be applied to cover the high malaria transmission period. This is consistent with current policy for SMC with SP plus amodiaquine in Burkina Faso where SMC is provided for four months.

In conclusion, the pharmacokinetic and pharmacodynamic properties of DHA-PQ were characterised during SMC in young children in Burkina Faso. Modelling and simulation predicted that, compared to the previously recommended dosage, the recently recommended increased dosage would reduce the malaria incidence by 38.8% in small children when administered as three months of SMC. Extending this therapy from three to four months of SMC with an increased dosage reduced the 120 days cumulative malaria incidence by 66.7%. If SMC with SP plus amodiaquine in West and Central Africa starts to lose efficacy because of drug resistance, DHA-PQ could be used as an alternative regimen. In parts of Southeast Africa, where transmission is highly seasonal but drug resistance to SP prevents the used of SMC with SP plus amodiaquine, DHA-PQ could also be used for SMC. Our study shows that if DHA-PQ is used for SMC, monthly treatment using the new dosing regimen should be substantially more effective in preventing malaria than the currently recommended regimen.

## Methods

**Study design and ethical approval**. This study was part of a randomised clinical trial to compare the protective efficacy of SP and amodiaquine and DHA-PQ for SMC in children at high risk of malaria, at three rural health facilities in the district of Lena located 40–50 km from Bobo-Dioulasso, the second largest city of Burkina Faso. This is an area of intense seasonal transmission with entomological inoculation rates during the dry and the rainy season months of 3.6 and 533 infective bites per person per year, respectively[42]. 754 children received DHA-PQ. A subset of 45 children was identified at randomisation for assessment of biochemical and haematological parameters, 15 children to be assessed after each of the SMC rounds in August, September, and October. A subset of 210 children in the DHA-

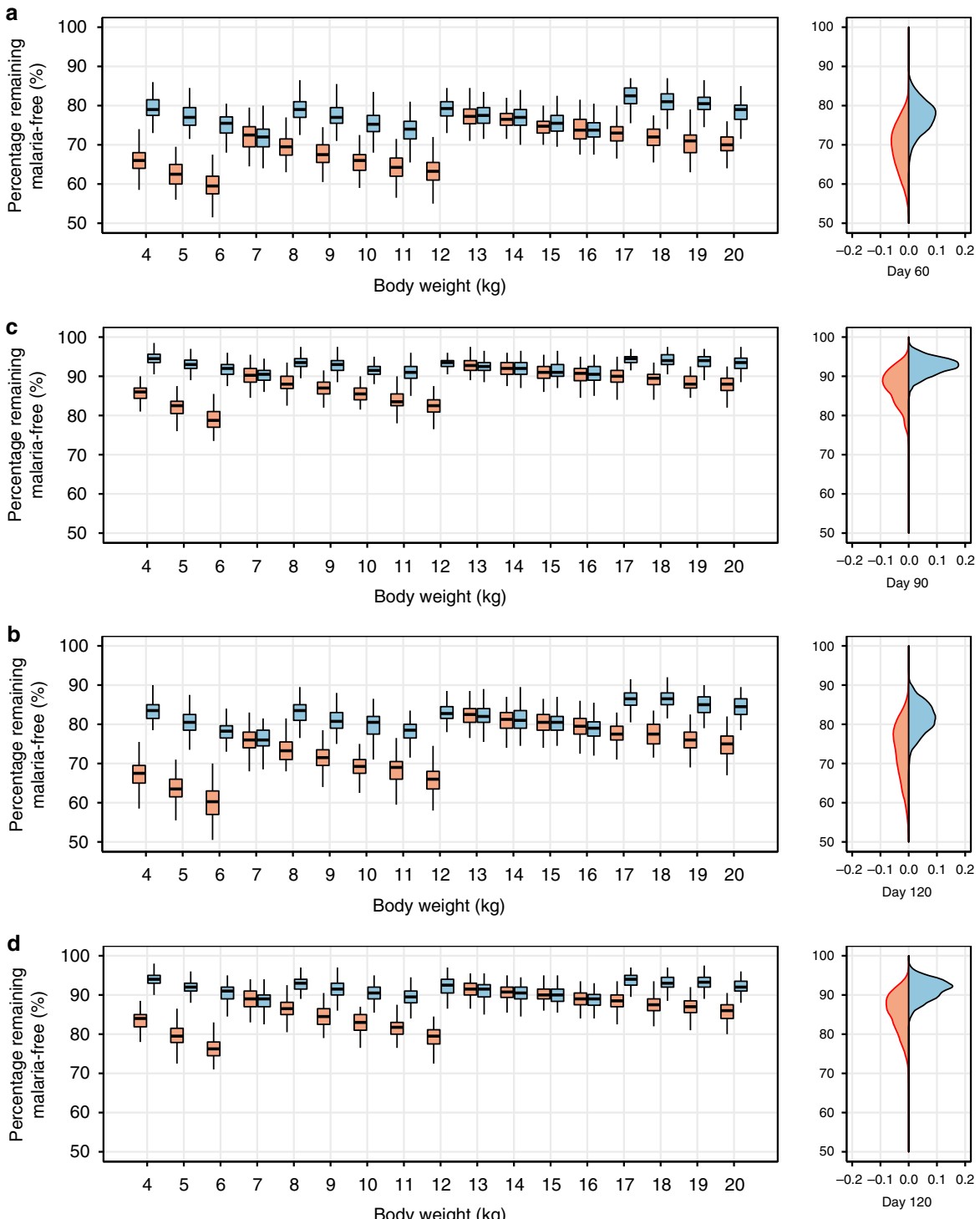

**Fig. 4** Comparisons of the expected pharmacodynamic outcomes. **a** Children (4–20 kg) remaining malaria-free after a single treatment regimen by day 60, **b** children (4–20 kg) remaining malaria-free after three rounds of a monthly dose regimen by day 120, **c** children (4–20 kg) remaining malaria-free after three rounds of a monthly dose regimen by day 90, **d** children (4–20 kg) remaining malaria-free after four rounds of a monthly dose regimen by day 120. Red box-whisker plots represent the standard WHO 2010 dosing regimen[52,43] and blue box-whisker plots represent the increased dosing regimens according to the revised WHO 2015 recommendation[26], stratified by body weight. The simulations are based on 200 individuals per body weight for 100 replications. The box-whisker plots represent the median with inter-quartile range and the 95% prediction interval. Right panels represent the proportions of children remaining malaria-free at the end time point; the red filled density plots represent the standard WHO 2010 dosing regimen and the blue filled density plots represent the revised WHO 2015 dosing regimen

**Table 3 Simulated malaria incidence in children given DHA-PQ seasonal malaria chemoprevention, following the WHO 2010 and 2015 dosing recommendations**

|  | Dosing regimen (WHO 2010) | Increased dosing regimen (WHO 2015) | Relative reduction (%) |
|---|---|---|---|
| Malaria incidence (%) at day 60, a single treatment | 33.0 (24.0–43.3) | 22.5 (17.5–34.5) | 32.3 |
| Malaria incidence (%) at day 120, three months of SMC | 27.0 (14.5–45.0) | 16.5 (9.00–26.0) | 38.8 |
| Malaria incidence (%) at day 90, three months of SMC | 12.0 (4.23–27.0) | 5.00 (1.50–10.5) | 58.3 |
| Malaria incidence (%) at day 120, four months of SMC | 13.0 (0.40–30.0) | 5.50 (1.50–11.5) | 57.6 |

All values are reported as median (95% confidence interval) unless otherwise specified
SMC seasonal malaria chemoprevention, WHO World Health Organization

PQ group was identified at randomisation to give blood samples for evaluation of the pharmacokinetic properties of piperaquine in the nested PKPD group of the study; 70 children were randomised to be sampled in August, 70 in September, and 70 in October. The children without blood samples were consider as a PD group (562 children) for external model validation. Individuals who provided insufficient information for pharmacokinetic modelling were omitted from this study. No pharmacokinetic data were collected in the SP and amodiaquine arm. Clinical details and results are reported in full elsewhere[6]. Study approval was obtained from the ethics committees of the London School of Hygiene and Tropical Medicine, United Kingdom, and the ethics committee of the Centre Muraz (Comité d'Ethique Institutionnel du Centre Muraz), Burkina Faso. The study was registered at www.clinicaltrial.gov (NCT00941785). The inclusion criteria were as follows: age between 3 and 59 months, expected to stay in the study area during the study period, no history of allergy to the study medication, and no chronic condition requiring hospitalization (e.g. severe malnutrition). The presence of malaria at enrolment was not an exclusion criterion; if malaria was diagnosed, the patient was enroled and treated with artemether-lumefantrine (Coartem®), and the SMC was not given, but the child was eligible to received subsequent monthly doses of SMC. The study was explained to the guardians in their own language, and written informed consent was obtained from all participants before enrolment. After initial enrolment, children returned to visit the study clinic monthly for clinical examination and for SMC drugs to be administered. Field workers visited the children at home for the next 2 days to supervise drug administration. The home visits were performed once every two weeks and any child who was unwell was referred to the clinic for assessment and care by a study physician.

**Drug regimens.** Children, 2.33–58.1 months of age, received a three-day fixed oral combination of DHA-PQ (Duocotexin®, Beijing Holley-Cotec Pharmaceutical, China, 40 mg dihydroartemisinin and 320 mg piperaquine tetra-phosphate per tablet) once a month for three consecutive months during the malaria seasonal (August – October, 2009). Dosing was administered according to standard weight-based treatment dosing guidelines, rounded to the nearest quarter tablet, with a target dose of 2 mg kg$^{-1}$ day$^{-1}$ of dihydroartemisinin and 18 mg kg$^{-1}$ day$^{-1}$ of piperaquine phosphate[43]. All drug administrations were supervised, and the date and time of administration was recorded. DHA-PQ was administered as whole tablets or fractions of tablets on an empty stomach with a glass of water. Tablets were crushed, mixed with water, and administered as a slurry to the children who could not ingest the tablets. Children were observed for 30 min after dosing to ensure that the medication was not vomited. Children who did vomit within 30 min of drug administration received an additional dose, and were subsequently excluded from the pharmacokinetic–pharmacodynamic analysis from that time point. All microscopy confirmed malaria cases were treated with a full treatment dose of artesunate-lumefantrine (Coartem®) and these children then excluded from further analysis from the time of treatment.

**Blood samples.** All children in the nested pharmacokinetic–pharmacodynamic study contributed a baseline finger prick capillary blood sample (~200 µl) before antimalarial treatment. One hundered and seventy-nine children were randomly allocated to be sampled during the three months of SMC (32 children were sampled in August, 92 children in September, and 55 children in October). Four finger prick samples were taken randomly from each child in pre-specified sampling windows (pre-dose, 0–6 days, day 7, and 8–30 days after the first day of dosing). One venous sample (~2 ml) was taken from 40 children at the same time as the second, third, or fourth finger prick sample. Finger prick samples and plasma samples were stored at −80 °C until shipment on dry ice to the Department of Clinical Pharmacology, Mahidol-Oxford Tropical Medicine Research Unit

(MORU), Bangkok, Thailand. Malaria was diagnosed using thick and thin blood smears, stained with 2% Giemsa for 30 min and examined in double readings by experienced laboratory technicians.

**Piperaquine quantification.** Piperaquine concentrations were measured using solid phase extraction followed by liquid chromatography coupled with tandem mass spectroscopy according to a previously published method[44]. Quality control samples at 4.5, 20, and 400 ng mL$^{-1}$ were analysed in triplicate within each batch of clinical samples to ensure the accuracy and precision of the assay. The relative standard deviation (%RSD) at low, middle, and high concentrations were 3.90%, 2.04%, 2.42% for venous samples and 5.29%, 4.51%, 3.69% for capillary samples. The limit of detection (LOD) and the LLOQ of both venous and capillary plasma samples were set to 0.375 and 1.50 ng mL$^{-1}$, respectively. The laboratory is a participant in the QA/QC proficiency testing programme supported by the Worldwide Antimalarial Resistance Network (WWARN)[45].

**Pharmacokinetic analysis.** Piperaquine capillary and venous plasma concentrations were transformed into their natural logarithms. Modelling and simulation was performed using NONMEM version 7.3 (Icon Development Solution, Ellicott City, MD). Piranha 2.9[46], and Perl-speaks-NONMEM (PsN; version 4.4.0)[47] were used for automation and diagnostics during the model building process. The concentration-time profile was characterised using non-linear mixed-effects modelling with the first-order conditional estimation method with interactions in NONMEM. The limited observed concentrations (~3–4 samples per individual) were not deemed sufficiently informative to characterise adequately the known multi-phasic pharmacokinetic profile of piperaquine. Typical pharmacokinetic parameter estimates and inter-individual variability estimates from a treatment study in children from the same region were therefore incorporated by using the $PRIOR functionality in NON-MEM[48]. The prior pharmacokinetic model was a flexible two-transit absorption compartment model followed by a three-compartment disposition model, including body weight as a fixed allometric function on all clearance and volume parameters[10]. A linear association between capillary and venous plasma concentrations was hypothesized and then modelled using an estimated conversion factor at the population level. Inter-individual variability of the pharmacokinetic parameters was assumed to be log-normally distributed with a zero mean and $\omega^2$ variance (Eq. 1). The unexplained residual variability was modelled separately for capillary and venous plasma concentrations and implemented as proportional error models on the log-transformed concentrations, essentially equivalent to proportional errors on an arithmetic scale. Body weight (BW) was introduced as an allometric function for all clearance (exponent of $n = 0.75$) and volume (exponent of $n = 1.00$) parameters, centralized to 18 kg of body weight according to the typical patient in the prior model (Eq. 1). A maturation process of enzyme-dependent metabolic elimination was evaluated based on the elimination clearance of piperaquine (Eq. 2)[49].

$$\theta_i = \theta \times \exp\left(\eta_{i,\theta}\right) \times \left(\frac{\text{BW}_i}{18.0}\right)^n \qquad (1)$$

$$\text{MF} = \frac{\text{PMA}^{\text{HILL}}}{\text{TM}_{50}^{\text{HILL}} + \text{PMA}^{\text{HILL}}} \qquad (2)$$

where $\theta_i$ is the individual pharmacokinetic parameter estimate of the $i^{\text{th}}$ subject, $\theta$ is the typical population parameter estimate, $\eta_{i,\theta}$ is the inter-individual variability, MF is the maturation factor, PMA is the post-menstrual age, HILL is the Hill coefficient, and $\text{TM}_{50}$ is the maturation half-time. Inter-study differences between the current study and the prior study were investigated by applying a categorical study covariate on all pharmacokinetic parameters. All other covariates (parasitaemia, gender, age,

and nutritional status) were investigated by a stepwise addition ($p < 0.05$) and elimination ($p < 0.01$) approach, using linear, exponential, and power functions.

**Pharmacodynamic analysis**. The final pharmacokinetic model and its typical parameter estimates were fixed, and the time interval to malaria infection during the follow-up period was modelled using a time-to-event analysis. Time to the first malaria episode or being lost/censored during five months of follow-up were used as primary outcome data for the pharmacodynamic model. Non-linear mixed-effects modelling with the Laplace estimation method with interactions was applied throughout the pharmacokinetic–pharmacodynamic modelling approach.

The protective effect of piperaquine ($PQ_{EFF}$, Eq. 3) was defined by a sigmoidal $E_{MAX}$ function, where $E_{MAX}$ is the maximal drug effect, $CP(t)$ is the predicted venous piperaquine plasma concentration at time $t$, $IC_{50}$ is the venous piperaquine plasma concentration needed to reduce the hazard of malaria infection by 50%, and $\gamma$ is a shape parameter. The effect of dihydroartemisinin ($DHA_{EFF}$) was implemented as a categorical function reducing the hazard of malaria infection to zero, six days before a treatment dose (see below) and during the three days of treatment in each period (Eq. 4). A blood stage infection was assumed to start at $\sim 10^5$ parasites emerging from the liver, and the microscopy detection limit of malaria was set to be approximately equal to a total parasite biomass of $10^8$ parasites. Thus, assuming an approximate 10-fold multiplication rate every 48 h in the absence of drug, a blood stage infection starting between 0 and 6 days before treatment would not be detected, since the exposure to dihydroartemisinin would eliminate such an infection before reaching a microscopy detectable parasite density.

$$PQ_{EFF} = 1 - \frac{E_{MAX} \times CP^{\gamma}(t)}{IC_{50}^{\gamma} + CP^{\gamma}(t)} \quad (3)$$

$$DHA_{EFF} = \begin{cases} 0 & \text{; from 6 days before the first dose to 24 hours after the last dose} \\ 1 & \text{; otherwise} \end{cases} \quad (4)$$

A constant baseline hazard of malaria infection during the high-transmission season was assumed, and the hazard function ($Hz(t)$) was defined by multiplication of the constant baseline hazard ($\theta_{BASE}$), piperaquine drug effect ($PQ_{EFF}$), and dihydroartemisinin drug effect ($DHA_{EFF}$), as illustrated in equation 5. The survival function was calculated as the exponent of the cumulative hazard (Eq. 6) and the probability density function for acquiring a malaria infection at a specific time point ($P(t)$) was therefore defined as the product of the hazard function with the survival function (Eq. 7).

$$Hz(t) = \theta_{BASE} \times PQ_{EFF} \times DHA_{EFF} \quad (5)$$

$$S(t) = \exp\left(-\int_0^t Hz(t)dt\right) \quad (6)$$

$$P(t) = S(t) \times Hz(t) \quad (7)$$

Different levels of parasitaemia at the time of malaria detection during the follow-up period indicated that the time of emerging blood stage infections might differ among subjects. The lag-time between emerging blood stage infections from the liver and the microscopy detection of parasites is dependent on several factors, i.e. parasite growth rate, number of parasites emerging from the liver, and drug concentration, as well as the quality of microscopy. The growth rate of malaria parasites is variable, particularly in a context of variable immunity, and is even more uncertain during the terminal elimination phase of a slowly eliminated antimalarial drug[33]. Therefore, the time interval before emergence of blood stage infection from the liver was back-extrapolated based on the observed number of detected parasites ($PAR_{OBS}$)[19,33,50]. To account for the uncertainties noted above, the start time of the interval ($I_{Start}$; Eq. 8) was approximated by assuming a relatively small number of blood stage parasites emerging from the liver ($\sim 10^4$ parasites) and a lower than maximum growth rate ($K_{Growth, slow}$; 5-fold increase every 48 h). The end time of the interval ($I_{End}$; Eq. 9) was approximated by assuming a relatively large number of blood stage parasites emerging from the liver (approximately $10^5$ parasites) and an unrestrained growth rate ($K_{Growth, fast}$; 10-fold increase every 48 h). This scheme is illustrated in Supplementary Figure 3. The final pharmacodynamic model was based on an interval-censoring time-to-event approach, accounting for the likely interval of emerging blood stage infections according to equations 10–11. Thus, the probability of an event occurring within this interval ($I_{Start} < t < I_{End}$) was determined by equation 10, as compared to the probability of an event occurring in patients with no detected parasitaemia during

follow-up time period ($T$) according to equation 11.

$$I_{Start} = \frac{1}{K_{Growth, slow}} \times \ln\left(\frac{PAR_{OBS}}{10^4}\right) \quad (8)$$

$$I_{End} = \frac{1}{K_{Growth, fast}} \times \ln\left(\frac{PAR_{OBS}}{10^5}\right) \quad (9)$$

$$P(I_{Start} < t < I_{End}|\theta) = 1 - [S(I_{End}) - S(I_{Start})] \quad (10)$$

$$P(t > T|\theta) = S(T) \quad (11)$$

Biologically plausible covariates (body weight, gender, and age) were evaluated with a stepwise approach as linear and exponential functions on baseline hazard in the final time-to-event model. Also, individually predicted venous plasma piperaquine drug concentrations at the back-extrapolated start of blood stage infections were estimated and summarised as conservative estimates of the MIC values for clinical infections, assuming very high-transmission intensity. Similarly, individually predicted piperaquine trough concentrations were assumed to be the maximum MIC values for patients who did not present with recurrent infections (i.e. realistic MIC values for patients without reinfections are likely to be lower). Thus, to avoid any potential bias towards lower target drug concentrations, the 95th percentile of individually predicted MIC values was calculated and presented as a clinical target MIC.

**Model diagnostics**. The pharmacokinetic and pharmacodynamic models were diagnosed by using both internal and external validation.

The objective function value (OFV), calculated by NONMEM as proportional to $-2 \times$ log-likelihood of data, was used to evaluate competing nested models. A reduction in OFV ($\Delta$OFV) is equivalent to a likelihood ratio test (LRT) and the $\Delta$OFV of 3.84, 6.63, and 10.8 was considered significant with a $p$-value of 0.05, 0.01, and 0.001, respectively, for two nested models with one degree of freedom difference. Shrinkage values were used to evaluate the reliability of the goodness-of-fit diagnostics, where high shrinkage values reflect a lower accuracy of individual predictions. A prediction-corrected visual predictive check ($n = 2000$ simulations) was applied to the final pharmacokinetic model in order to determine the predictive power of the model on the low (5th), median (50th), and high (95th) percentiles[51]. A visual predictive check of the time-to-event model was obtained by visualising the observed Kaplan–Meier plot with the 95% confidence interval of simulated events ($n = 1000$). As external validation, the final time-to-event model and the demographic and dosing information from the larger PD group ($n = 562$) were used to simulate 1000 new clinical studies. The simulated times to malaria infection were overlaid with the observed incidence of malaria in the same group. Non-parametric relative standard errors (%RSE) and model robustness were assessed by bootstrapping diagnostics ($n = 500$). All diagnostic procedures were implemented by using PsN.

**Pharmacokinetic–pharmacodynamic outcome simulations**. The final pharmacokinetic–pharmacodynamic model was used to simulate the antimalarial protective effect of DHA-PQ after different dosing regimens (Supplementary Table 2). Pharmacokinetic concentration-time profiles of 1000 children at each kg body weight (4–20 kg) were simulated and day 7 concentrations and peak piperaquine concentrations were summarised and compared between dosing strategies. The final time-to-event model was used to simulate the therapeutic outcome, stratified by body weight ($n = 200$ individuals per body weight, 100 replicates), in order to compare the malaria incidence between children receiving the standard WHO 2010 regimen[52,43] and children receiving the WHO 2015 suggested increased dosing regimen, based on Tarning et al.[10,26]. The therapeutic outcomes of a single treatment, three months of SMC, and four months of SMC were simulated and compared in order to determine the optimal dosing scenario.

**Reporting summary**. Further information on experimental design is available in the Nature Research Reporting Summary linked to this article.

## Data availability
All relevant data and NONMEM code for the pharmacokinetic and pharmacodynamic model are available from the authors upon reasonable request.

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

## Acknowledgements

We thank all children and their parents for their participation in completing this study. We also thank the diligent staff from Institut de Recherche en Sciences de la Santé, Burkina Faso. This investigation was supported by Beijing Holley-Cotec Pharmaceuticals and the London School of Hygiene and Tropical Medicine. The study was a part of the Wellcome Trust-Mahidol University-Oxford Tropical Medicine Research Programme supported by the Wellcome Trust of Great Britain. The Bill and Melinda Gates Foundation also supported part of this work. IZ was supported by a fellowship from the IAEA.

## Author contributions

I.Z., P.M., Y.D.C., A.F.S., D.C., F.N., B.G., P.J.R., N.J.W., J.B.O. and J.T. conceived the project. W.H. quantified the drug concentrations. P.C. and J.T. performed the pharmacokinetic–pharmacodynamic analysis and wrote the first draft of the manuscript. All authors revised and approved the final manuscript.

## Additional information

**Competing interests:** The authors declare no competing interests.

