## [Peer Review File · Nature Communications]

Reviewers' Comments:

Reviewer #1:

Remarks to the Author:

The manuscript describes an important effort to improve anti-malaria treatment in young children and will benefit from clarifying the following points prior to being considered for publication:

1. The manuscript needs thorough editorial works. Abbreviations are not defined at its first use and are not consistently used through the manuscript (i.e., DHA-PQ). In addition, inconsistent terms are used to describe a same thing (i.e., PK group vs PK-PD group, PD group vs. PD safety group). Lastly, some results are mixed in Methods or Discussion (such as "In practice, 179/210 children gave blood samples, 32, 92 and 55 respectively in August, September and October", Children, 2.33-58.1 months of age, received a three-day fixed...) or vice versa (introducing Bergstrand et al. [15] model in Methods rather than Results).

2. Abstract:

- Add "piperazine" when the abstract mentions "The use of dihydroartemisinin as an alternative...".
- Body weight was not tested for statistical significance, but was included with fixed 1 for V and 0.75 for CL. Please clarify.
- Suggest clarifying "the higher dose" is WHO 2015 guideline dosing regimen.

3. Introduction:

- Please add the known PK properties of PQ in adults and children (if any), including the absorption (food effect), distribution and metabolism/elimination pathways. This will greatly help later some of the modelling decisions (i.e. why maturation process of enzyme-dependent metabolic elimination) was introduced.
- I think some general description of Reference 6 (the original study), including key I/E criteria, is useful to add.

4. Results:

- Results need to be clarified that the PK and PK-PD analyses were primarily based on n=179 while n=562 were used as an external validation. Please clarify to focus on 179 subject data first. In addition, Table 1 has more than just demographic data and needs further description in text to include important data aspects, such as 466/71 PK samples from children 2.33 to 58.1 months.
- Table 1 needs to clarify that those children without blood samples consider as a pharmacodynamic group (562 children) were used for external model validation.
- Table 1 shows average 29 mg/kg/month while the Method states that the target was 18 mg/kg/day (which translate to 54 mg/kg/month). Please discuss this discrepancy in Discussion.
- For PK analysis, no covariate testing was done. Please include the rationale in Discussion.
- Considering that inclusion of MF with PMA improved the model fit significantly for those < 2 years and the TM50 was ~1 month when the youngest subject was 2.3 months old, suggest testing a simpler MF model with age (i.e. power function). Please add how such alternative models were considered before omitting maturation effect in the final model.
- Suggest moving the diagnostic plots in Figure 1 to supplementary.
- Please clarify observed day 7 concentration in Results and Table S1. Is it a trough or average or at a random time during Day 7?

5. Discussion

- Please include why capillary PQ concentrations were higher than venous measurements.
- Please discuss limitations of the current work (sparse sampling, no PK of DHA, assumptions, etc...) and suggest adding some discussion/references to support some key assumptions in the modelling to

help readers who are not familiar with the therapeutic area:

- o "assuming an approximate 10-fold multiplication rate every 48 hours in the absence of drug"
- o "A blood stage infection was assumed to start at approximately 105 parasites emerging from the liver, and the microscopy detection limit of malaria was set to be approximately equal to a total parasite biomass of 108 parasites"
- o "assumes that the full 3-day treatment would completely eliminate circulating parasites"

Reviewer #2:

Remarks to the Author:

Dr. Tarning and co-authors have provided a well written and important manuscript describing the PK of piperazine in pediatric subjects, and the PK/PD of dihydroartemisinin-piperazine regimens for seasonal malaria chemoprevention (SMC) in young children in areas of high levels of malaria incidence.

The primary result of this work is the provision of new clinical data that led to modeling/simulation based evidence to refine the dosage regimen for SMC in small children. The work supports increased dosages of piperazine in small children and the extension of the treatment period from three to four months. WHO Guidelines were updated in 2015 to recommend increased treatment doses of piperazine for small children. The present work extends that recommendation to cover SMC therapy, and to a recommended extension of SMC as well.

The methods of the trial are valid for the goals of the investigation. Children in a region of intense seasonal transmission (Burkina Faso) were administered SMC therapy. Proper ethical procedures were described. Standard clinical weight-based doses were administered for 3 days in each of 3 months. Microscopically confirmed cases of malaria were treated with Coartem as a rescue medication and excluded from further analysis. Upon development of the PK/PD model additional dosing regimens were explored using modeling and simulation. The data were useful for model development followed by simulation to explore the impact of different dosage regimens.

The work is of clinical importance in that evidence based decisions are informed that may potentially reduce the incidence of malaria in small children substantially. The evidence of the effects of the alternate dosage regimens is based on modeling and simulation and could be confirmed with clinical data in future studies.

The data collected for the trial were reasonable for the investigation. Over 700 children were studied, with PK/PD samples collected for a portion of the children (179, the model development group) and the pharmacodynamic group consisted of 562 children (used for external validation, not model development). Assay methodology results are presented and are sufficient.

Appropriate statistical methods have been used throughout the work. Several challenges needed to be addressed with modeling/statistical approaches. The use of frequentist priors to support the PopPK model parameter estimation appropriately addresses the sparseness of the data within subjects. The correlation of capillary and venous drug concentrations appropriately allows for PK parameter estimation using the combined data from both collection methods. Numerical predictive checks and bootstrap assessments supported the appropriateness of the model developed for the observed data. Visual predictive checks also support the time-to-event model for occurrence of recrudescence.

In line 379 to 380, I do question if the text providing interpretation of the residual variability form is correct. I believe an additive residual error in the log-domain would be essentially equivalent to the

proportional error in the additive domain; not the exponential error in the log-domain as is stated in line 380.

Another question for clarification. In line 396, were the final PK model parameters that were fixed, set to the typical value or individual Bayesian values. I assume that due to the high degree of shrinkage, these are the typical values, but clarification would be useful.

In line 447/equation 11, definition of "T" might be useful.

Line 479: "individual" should be "individuals".

The conclusion of the manuscript does seem to be well supported and appropriate. The manuscript is very clearly written and supported by a well referenced introduction that sets the current work in the context of malaria treatment in general and SMC in particular. The role of SMC is also discussed in the context of the development of resistance to anti-malaria drugs.

Reviewers' comments:

Reviewer #1 (Remarks to the Author):

The manuscript describes an important effort to improve anti-malaria treatment in young children and will benefit from clarifying the following points prior to being considered for publication:

1. The manuscript needs thorough editorial works. Abbreviations are not defined at its first use and are not consistently used through the manuscript (i.e., DHA-PQ). In addition, inconsistent terms are used to describe a same thing (i.e., PK group vs PK-PD group, PD group vs. PD safety group). Lastly, some results are mixed in Methods or Discussion (such as “In practice, 179/210 children gave blood samples, 32, 92 and 55 respectively in August, September and October”, Children, 2.33-58.1 months of age, received a three-day fixed...”) or vice versa (introducing Bergstrand et al. [15] model in Methods rather than Results).

Thank you for these comments. We have gone through the manuscript for editorial review and we hope that it is now consistent and clear. Specifically;

- The term “DHA-PQ” have been used thorough the manuscript.
- The term “PKPD group” and “PD group” have been used thorough the manuscript.
- The sentence (“In practice, 179/210 children ...”) was removed from the Methods.
- We do think that “Children, 2.33 – 58.1 months of age, received a ...” belongs to the method section considering that it describes the study design.
- The PD model developed in this study was built on previous work form Bergstrand et al. [15], and we believe that it is important to acknowledge this in the method section while describing the modelling framework.

2. Abstract:

- Add “piperazine” when the abstract mentions “The use of dihydroartemisinin as an alternative...”.

Changed accordingly.

- Body weight was not tested for statistical significance, but was included with fixed 1 for V and 0.75 for CL. Please clarify.

Allometrically scaled bodyweight on the pharmacokinetic parameters are commonly included without formal statistical testing, considering the strong biological reasoning for this (e.g. a small child will undeniably have very different absolute capacity for drug metabolism compared to a large adult; hence mg/kg dosing). This is especially important when studying children, resulting in several fold differences in body weight between study participants. Furthermore, this covariate was included in the prior model that was applied and therefore an integral part of the model (and the original work showed a drop in OFV of -45 [$p < 0.001$] when evaluating this covariate formally).

It was clarified in the method section that this covariate relationship was already included in the prior model.

- Suggest clarifying “the higher dose” is WHO 2015 guideline dosing regimen.

Added accordingly.

3. Introduction:

- Please add the known PK properties of PQ in adults and children (if any), including the absorption (food effect), distribution and metabolism/elimination pathways. This will greatly help later some of the modelling decisions (i.e. why maturation process of enzyme-dependent metabolic elimination) was introduced.

Thank you for this suggestion. We have added this information in the introduction as requested.

- I think some general description of Reference 6 (the original study), including key I/E criteria, is useful to add.

Some general information on study design, including inclusion/exclusion criteria have been added in the first section of the method (Study design and ethical approval).

4. Results:

- Results need to be clarified that the PK and PK-PD analyses were primarily based on n=179 while n=562 were used as an external validation. Please clarify to focus on 179 subject data first. In addition, Table 1 has more than just demographic data and needs further description in text to include important data aspects, such as 466/71 PK samples from children 2.33 to 58.1 months.

Thank you for these comments. The results have been clarified with respect to number of children included in the primary analysis, and those used for external validation. In addition, Table 1 has been described further.

- Table 1 needs to clarify that those children without blood samples consider as a pharmacodynamic group (562 children) were used for external model validation.

Changed accordingly.

- Table 1 shows average 29 mg/kg/month while the Method states that the target was 18 mg/kg/day (which translate to 54 mg/kg/month). Please discuss this discrepancy in Discussion.

Thank you for point out this apparent discrepancy. Table 1 show the average dose of piperazine base while the recommendation is made with respect to piperazine phosphate (conversion factor of approximately 0.5). Thus, the children in this study was dosed according to the recommendations, and we have clarified this by reporting piperazine phosphate in Table 1.

- For PK analysis, no covariate testing was done. Please include the rationale in Discussion.

Apologies for this confusion. The current study did perform a formal covariate analysis, as stated in the material and methods “All other covariates were investigated by a stepwise addition ($p < 0.05$) and elimination ($p < 0.01$) approach.” To avoid the confusion, similar text was added in the results section.

- Considering that inclusion of MF with PMA improved the model fit significantly for those < 2 years and the TM50 was ~ 1 month when the youngest subject was 2.3 months old, suggest testing a simpler MF model with age (i.e. power function). Please add how such alternative models were considered before omitting maturation effect in the final model.

Thank you for this insightful comment. Age was evaluated as an independent covariate, using linear, exponential, and power functions, during the SCM procedure. However, it was not statistically significant in any of these models. We have modified the method section (covariate modelling) to clarify this approach.

- Suggest moving the diagnostic plots in Figure 1 to supplementary.

Moved accordingly.

- Please clarify observed day 7 concentration in Results and Table S1. Is it a trough or average or at a random time during Day 7?

Observed day 7 concentrations were all samples collected between 166 and 171 hours after the first dose. Note that only one sample was collected per patient on this day, and that collection time varied depending on when the patient came to the clinic. However, the day 7 concentrations in Table S1 are individually predicted day 7 concentrations, i.e. at 168 hours after the first dose.

5. Discussion

- Please include why capillary PQ concentrations were higher than venous measurements.

We apologize for this omission. Text has been added to clarify this phenomenon (which is seen in all studies evaluating piperazine concentrations in different biological matrices).

- Please discuss limitations of the current work (sparse sampling, no PK of DHA, assumptions, etc...) and suggest adding some discussion/references to support some key assumptions in the modelling to help readers who are not familiar with the therapeutic area:
 - o “assuming an approximate 10-fold multiplication rate every 48 hours in the absence of drug”
 - o “A blood stage infection was assumed to start at approximately 105 parasites emerging from the liver, and the microscopy detection limit of malaria was set to be approximately equal to a total parasite biomass of 108 parasites”
 - o “assumes that the full 3-day treatment would completely eliminate circulating parasites”

Thank you for these comments. The limitations have been expanded to also include the issues mentioned above.

Reviewer #2 (Remarks to the Author):

Dr. Tarning and co-authors have provided a well written and important manuscript describing the PK of piperazine in pediatric subjects, and the PK/PD of dihydroartemisinin-piperazine regimens for seasonal malaria chemoprevention (SMC) in young children in areas of high levels of malaria incidence.

The primary result of this work is the provision of new clinical data that led to modeling/simulation based evidence to refine the dosage regimen for SMC in small children. The work supports increased dosages of piperazine in small children and the extension of the treatment period from three to four months. WHO Guidelines were updated in 2015 to recommend increased treatment doses of piperazine for small children. The present work extends that recommendation to cover SMC therapy, and to a recommended extension of SMC as well.

The methods of the trial are valid for the goals of the investigation. Children in a region of intense seasonal transmission (Burkina Faso) were administered SMC therapy. Proper ethical procedures were described. Standard clinical weight-based doses were administered for 3 days in each of 3 months. Microscopically confirmed cases of malaria were treated with Coartem as a rescue medication and excluded from further analysis. Upon development of the PK/PD model additional dosing regimens were explored using modeling and simulation. The data were useful for model development followed by simulation to explore the impact of different dosage regimens.

The work is of clinical importance in that evidence based decisions are informed that may potentially reduce the incidence of malaria in small children substantially. The evidence of the effects of the alternate dosage regimens is based on modeling and simulation and could be confirmed with clinical data in future studies.

The data collected for the trial were reasonable for the investigation. Over 700 children were studied, with PK/PD samples collected for a portion of the children (179, the model development group) and the pharmacodynamic group consisted of 562 children (used for external validation, not model development). Assay methodology results are presented and are sufficient.

Appropriate statistical methods have been used throughout the work. Several challenges needed to be addressed with modeling/statistical approaches. The use of frequentist priors to support the PopPK model parameter estimation appropriately addresses the sparseness of the data within subjects. The correlation of capillary and venous drug concentrations appropriately allows for PK parameter estimation using the combined data from both collection methods. Numerical predictive checks and bootstrap assessments supported the appropriateness of the model developed for the observed data. Visual predictive checks also support the time-to-event model for occurrence of recrudescence.

Thank you for these positive comments regarding the manuscript.

In line 379 to 380, I do question if the text providing interpretation of the residual variability form is correct. I believe an additive residual error in the log-domain would be essentially equivalent to the proportional error in the additive domain; not the exponential error in the log-domain as is stated in line 380.

Thank you for point out this mistake. This has been edited accordingly.

Another question for clarification. In line 396, were the final PK model parameters that were fixed, set to the typical value or individual Bayesian values. I assume that due to the high degree of shrinkage, these are the typical values, but clarification would be useful.

Edited accordingly.

In line 447/equation 11, definition of "T" might be useful.

Edited accordingly.

Line 479: "individual" should be "individuals".

Edited accordingly.

The conclusion of the manuscript does seem to be well supported and appropriate. The manuscript is very clearly written and supported by a well referenced introduction that sets the current work in the context of malaria treatment in general and SMC in particular. The role of SMC is also discussed in the context of the development of resistance to anti-malaria drugs.

Reviewers' Comments:

Reviewer #1:

Remarks to the Author:

My comments were addressed.